# Developing undergraduate autism education for medical students: a qualitative study

Yasmin Dhuga,[1] Yvonne Feeney,[1] Laura Gallaher,[2] Ann White,[3] Juliet Wright,[1] Sube Banerjee,[4] Stephanie Daley  [1]

[1]Brighton and Sussex Medical School, University of Sussex, Brighton, UK
[2]School of Media Arts and Humanities, University of Sussex, Brighton, UK
[3]Sea Side View, Child Development Centre, Sussex Community NHS Foundation Trust, Brighton, UK
[4]Faculty of Health and Human Sciences, University of Plymouth, Plymouth, UK

**Correspondence to**
Dr Stephanie Daley; s.daley@bsms.ac.uk

## ABSTRACT

**Background** Autistic adults and children experience considerable health inequalities and have high rates of premature mortality, hospital admissions and emergency department visits. This is in part due to a lack of autism awareness in the healthcare and social care workforce. A new educational programme, Time for Autism (TfA), for medical students is being developed to address this challenge. This qualitative study was undertaken to support the development of the new programme in order to (1) understand the medical care experiences of parents of autistic children and (2) assess their views on the acceptability of the new TfA programme and willingness to be involved.

**Methods** A convenience sample of 11 parents of autistic children were recruited across the South of England. The ages of the autistic children ranged from 3 to 17 years. Semistructured interviews were completed between October and December 2019. Interview transcripts were analysed using thematic analysis.

**Results** Three key themes were identified: diagnosis, experiences of doctors and TfA considerations. There was support for and willingness to take part in a dedicated autism education programme for medical students, and constructive feedback to inform and improve its delivery.

**Conclusion** The findings from this study provide insights into the medical care experiences of parents/carers of autistic children. Understanding how parents/carers of autistic children would like medical care to be improved can be used to develop TfA and other autism programmes. Parental/carer support for the development of and involvement in an autism medical education programme enhances the feasibility of the new programme.

## BACKGROUND

Autism is a multifaceted condition affecting neurodevelopment. It is characterised by impaired social interaction and communication, restricted interests and repetitive, stereotypical behaviours.[1] There are over 700 000 autistic people in the UK,[2] with prevalence increasing.[3] The cause of increasing prevalence is unknown but likely to be due in part to increased community awareness among the general population, including families and teachers and progress in case identification and definition.[4]

### WHAT IS ALREADY KNOWN ON THIS TOPIC

⇒ Parents/children of autistic children have high levels of dissatisfaction with the medical care provided to their children.
⇒ There is a lack of autism knowledge in doctors which contributes towards this problem.
⇒ For medical students, there is a lack of evidence as to how this knowledge gap can be addressed.

### WHAT THIS STUDY ADDS

⇒ The identification of areas of practice in which parents/carers perceive that autism education should be addressed in medical students.
⇒ The willingness of parents/carers to contribute their lived experience within autism education to address these concerns.
⇒ The introduction of a new autism education model, Time for Autism, for medical students in which lived experience is a core component.

### HOW THIS STUDY MIGHT AFFECT RESEARCH, PRACTICE OR POLICY

⇒ This study introduces a novel model of introducing autism educaiton into curriculum for medical students, and potentially other healthcare professionals in training.

Autistic people, both adults and children, are more likely to have additional health conditions compared with the general population,[5] including both mental health problems such as anxiety-related disorders[6] and sleep disorders,[7] as well as physical health problems such as gastrointestinal conditions.[8] Autistic adults and children are also at greater risk of health inequalities, and many autistic adults report negative healthcare experiences.[9 10] Parents of autistic children describe the healthcare management of their children as inadequate.[11] It is widely recognised that poor care experiences, as well as serious concerns about the quality of care provided to autistic adults and children, demonstrate a need for improved understanding of autism

across the life span within the health and social care workforce.[12–16]

Medical doctors are expected to have appropriate skills, competencies and understanding to deliver high-quality care to autistic adults and children[14] In England, a competence framework for working with autistic people across the life span has been developed and embedded within statute for the existing health and social care workforce.[17] However, at the undergraduate medical curriculum level, a time when attitudes and openness to learning can be most malleable, autism teaching provision is often patchy and inconsistent. To enable the future generation of doctors to fully meet the needs of autistic adults and children and deliver care in a way that is person-centred, autism education needs to be routinely embedded in the undergraduate medical curricula.[12 18] There is a gap in the literature about the efficacy of successful interventions that can enhance student understanding of autism.

While the literature on autism education for medical students is limited, there is some evidence indicating that interventions incorporating lived experience can have a positive impact. For example, one study reported an improvement in knowledge and understanding in third year medical students that completed panel discussions with autistic people.[19] In another study, the use of a 'digistory' to share the experiences of parents/carers of autistic children led to medical students reporting greater empathy, improved confidence and an ability to challenge negative stereotypes towards autism.[20]

The incorporation of lived experience to raise awareness and enhance skills towards autism in undergraduate medical students is a key feature of a new autism education programme that is being developed at Brighton and Sussex Medical School. The Time for Autism (TfA) programme will form a mandatory component of the undergraduate medical curriculum. The programme is modelled on a similar educational programme[21 22] that has successfully enhanced understanding and attitudes of dementia in large cohorts of healthcare students.[22 23] In TfA, pairs of medical students will visit a family (parent/carer with an autistic child) on three occasions over one academic year. Students will learn about the lived experience of autism in the context of authentic relationship built between the family and student learners. It is intended that the longitudinal nature of the programme will allow students sufficient time to develop a fuller understanding of how autism affects the child and family over time, further facilitating improved understanding. Supporting teaching activities will focus on autism across the life span, not just in childhood, and it is anticipated that several of the parents taking part in the new programme will also have a diagnosis of autism, further enhancing the learning for students.

Critical to the establishment of TfA will be the recruitment of volunteer families willing to have medical students visit them at home for the purpose of the new programme. To assess feasibility and inform the development of TfA, a qualitative study was carried out with parents/carers of autistic children to understand (1) their experiences of medical care towards their autistic child and (2) their views about the acceptability, value and potential willingness to get involved in the TfA.

## METHODS

### Study design

We carried out a qualitative study using individual semis-tructured interviews to explore parent/carer views about the medical care experiences of their autistic children as well as a new medical education programme in more depth.

The study considered the *reality* of participants' perceptions through an exploration of their experiences and interpretations they attach to them, within a broader social *construct*.[24] This epistemological approach loaned itself to the phenomenological methodology of thematic analysis and is steered by Braun and Clarke's guided approach to thematic analysis.[25]

### Setting and participants

A convenience sample of participants were recruited. Eligible participants were parents/carers of an autistic child under the age of 18 years living in the South of England. Participants were recruited from local autism charities, local NHS trusts, and online parent forums and they could self-refer to participate in the study. Participants were approached by email by the researchers and were not known to the research team before interview. Written informed consent was obtained from all participants.

Eleven participants took part in this study. Sociodemographic characteristics are shown in table 1.

**Table 1**  Sociodemographic characteristics

| Baseline characteristics | n | % Mean (SD) | Range (years) |
|---|---|---|---|
| Age | | 43.64 (7.75) | 29–55 |
| Age of autistic child | | 10.18 (4.73) | 3–17 |
| Ethnicity | | | |
| White background | 10 | 90.9 | |
| Mixed background | 1 | 9.1 | |
| Relationship to child | | | |
| Mother | 10 | 90.9 | |
| Father | 1 | 9.1 | |
| Marital status | | | |
| Married/partnered/cohabiting | 11 | 100 | |
| Highest educational level | | | |
| No qualification | 1 | 9.1 | |
| High school/college | 6 | 54.5 | |
| University or postgraduate degree | 4 | 36.4 | |
| Employment | | | |
| Unemployed | 1 | 9.1 | |
| Employed | 10 | 90.9 | |

There was no specific lived experience advisory involvement in this study, in relation to the study design, delivery and analysis, although this has since been established for the TfA programme.

## Procedure

Interviews were carried out by two researchers, SD and LG. SD is an experienced qualitative postdoctoral researcher who trained and supervised LG. Each interview was completed at a time and venue suitable to participants; the majority took place at home with only the participant present. Interviews were undertaken on one occasion and were audio recorded and lasted between 30 and 60 minutes. A topic guide was reviewed by two autism clinicians to ensure suitability and interview questions focused on the experience of the diagnosis of autism, experiences of consulting doctors (generally) about their autistic child, views about the TfA programme and willingness to participate. A pilot interview did not take place. Transcripts were not returned to participants, and feedback on findings was not sought. At the time of undertaking this study, the TfA programme was at an early stage of its development, and we did not have an established lived experience advisory group to call on to help us to interpret the interview findings.

## Data analysis

Interviews were transcribed verbatim and checked for accuracy. Transcripts were anonymised, and identifiable data were removed to maintain confidentiality. Thematic saturation was achieved after 11 interviews were completed; therefore, data collection stopped at this point.[26]

Transcripts were analysed using inductive thematic analysis.[25] Familiarity with the data was achieved by reading the transcripts multiple times. To ensure rigour, two separate researchers (YD and YF) independently coded two transcripts, before meeting to discuss the initial codes and resolve disagreements. Fieldwork diaries were also reviewed. The remaining transcripts were independently coded by the lead author, adding more codes when further meaningful data arose. The two researchers worked together to identify relationships between the codes and establish the main themes from the data. The themes were reviewed then refined and appropriate names were assigned to each theme. Microsoft Excel was used to aid the collation of the data.

To enhance the analysis of data, researchers kept fieldwork diaries and made notes after each interview. In addressing reflexivity, the researchers met regularly to discuss their positionality and its impact on the findings during data collection and analysis.

## RESULTS

Three main themes were identified from the data: diagnosis, experiences of doctors and TfA considerations.

These themes are summarised in table 2 and discussed further.

## Diagnosis

Participants reported that the diagnosis of autism was usually provided by either the child's general practitioner (GP) or their paediatrician. The age of diagnosis ranged from 3 to 15 years. This theme captured both positive and negative experiences reported by the participants during the diagnosis of their child.

Positive experiences of diagnosis were influenced by doctors that were non-judgemental, and empathetic about the challenges which participants faced prior to diagnosis. Participants who felt listened to by medical professionals reported better experiences during the diagnosis of their child.

However, participants reported more negative than positive experiences of doctors relating to the diagnosis of autism in their child. The delay in obtaining a diagnosis and the process of obtaining a diagnosis were often problematic for participants. Several explanations were provided to account for this, including (1) a perceived lack of autism awareness, (2) a perception that parents' concerns were not being taken seriously and (3) perceived difficulty by GPs in identifying symptoms of autism. The presence of comorbid symptoms was perceived to contribute to the challenges obtaining a diagnosis; for example, anxiety was not always viewed as an autistic trait. Another recurrent problem reported was a belief that GPs did not always use the correct diagnostic methods, for example, not using standardised tools to enable them to screen for or to diagnose autism, which delayed diagnosis further.

## Experiences of doctors

Participants discussed the general experiences they encountered with the wider range of doctors treating their autistic child.

Positive experience related to doctors who took a holistic approach and demonstrated patience and allowed the autistic child time to adapt to new environments. Using personalised approaches and engaging with the child's interests before discussing their wider health issues were appreciated. Participants valued doctors who recognised that autistic children have different needs and those who took their time to listen and acknowledge parents/carers and their concerns.

Equally, participants discussed many negative experiences of doctors. Most commonly, this related to a perceived lack of autism awareness and experience in doctors.

Some participants described a perceived arrogance in doctors and a failure to listen to the parent/carer autism-related concerns about their child. As a result, participants suggested that doctors did not adjust their practice to meet the needs of their child, causing additional stress for the parent/carers. Some participants felt that GPs, in particular, dismissed their concerns when their children

**Table 2** Participant quotations

| Theme/subtheme | Quotation |
|---|---|
| Diagnosis | Dr X who is just amazing, … she knows all about masking and everything… So, we had the assessment, Dr X listened to what we were saying, and actually listened.(Participant 9)<br>This is our GP [responding to parent concerns]… "Well, he's not ripping up my surgery, so there can't be much wrong with him" and I just felt like putting my head in my hands, because I just thought …<br>This is going to be a battle isn't it? (Participant 1)<br>So, again they [the gp] kind of felt …, it was just anxiety, it was kind of anxiety was the kind of prevailing sort of presentation [and not autism as the parent suspected and was later confirmed] and … they did the CBT approach. (Participant 7) |
| Experiences of doctors | …he [the gp] sort of crouched down…, sort of put his elbows on the bed, and was like really interested in her [autistic child]. (He) didn't talk to us [parents] first, he talked to her first…So, that was a really good example. (Participant 4)<br>…[autistic child] wouldn't engage with the doctor, he wouldn't go, so, I rang the doctor, she said "Come and see me" without him [child], to talk about him, which was great, because I was able to go. And she gave me as much time in the world to talk about him and his difficulties and what was going on at school,. And she said, we need to build up his tolerance to come in to visit the doctor. So, she said "Tell you what, next time, make an appointment with me… and if he won't get out the car, I'll come out and see him. (Participant 9)<br>the first GP that I saw … I asked at that time for …a referral psychology around his eating, I felt that they just kind of brushed that off and said that "Well, he's too young and it'll kind of sort itself out", but obviously we had several years of quite a restricted eating, So, yeah, so it would have been really good if the GP had kind of spent a bit more time and taken me seriously. (Participant 7) |
|  | [medical team] are relentlessly dismissive and… (Participant 1)<br>What would surprise me and make me feel better, is if they were like "Okay, excellent! What's the best way for me to do this with her? Or I need to do this, these are the things I need to do, how's the best way to do it? (Participant 2) |
| TfA considerations | I think it's brilliant … autism is something that is not well understood … I didn't realise how varied it is and how it displays itself really differently; my child is so, I think it's brilliant to have something that will help health professionals, or doctors in this case, understand a bit more… So, I think it's great and I really support it. (Participant 2)<br>… some of the frustrations that we've experienced really, have been where people have had a very medicalised mechanist understanding of what autism is and … something that is a problem to be fixed,. And so, I think one of the best ways I can think of for new doctors to get a different understanding of it is just to spend time around it, exactly as the programme is proposing. So, I think in that respect that's great. (Participant 6)<br><br>… it would just have to make it like a home visit outside of school time, and don't be late. We can't be that flexible, once the time is booked don't change it unless you have to, because that would then probably create some anxiety … but that would just be more because if we've done a lot of work around who's coming [to the house] when… (Participant 4) |

become physically unwell, resulting in avoidance of primary care by parents/carers and use of use of Emergency Department as a first step in receiving medical care. Overall, participants concluded that doctors see autism as a 'problem'.

Participants expected that all doctors should have an appropriate understanding about autism and a willingness to involve parent/carers in decision making about their autistic child's care. Participants felt that doctors should involve parents/carers early in the consultation so they can guide the doctor on how best to lead the consultation. Participants highlighted that as autism manifests differently, doctors should find out what works for each individual autistic child by asking the person who knows them best, their parent. Practical suggestions included doctors physically lowering themselves to the child's level and the use of easily understood language, rather than jargon.

Participants also suggested that doctors should be willing to deviate from routine established practice when assessing an autistic child because they may present atypically or the child may be unwilling to engage, for example, may be willing to assess the child in the family car if the child would not come into the GP practice.

**TfA considerations**
This theme focused on participant's views about the TfA programme and potential willingness to take part. Two

subthemes were identified: positive attitudes about TfA and visit considerations.

**Positive attitudes about TfA**
All participants were positive about the development of the TfA programme and the potential for it to address the perceived lack of autism awareness in doctors. Participants stated that longitudinal contact with children in the community would enhance medical student understanding of autism and thereby improve their future practice. Participants suggested that gaining direct experience with autistic children would be more valuable than learning about autism from a textbook. Participants also suggested that medical students would be able to learn more about the impact of autism on the family, potentially enabling doctors to better manage difficult family situations. Finally, all participants reported their autistic children would also enjoy being part of TfA.

**Visit considerations**
All participants were keen for TfA to take place and provided practical recommendations to support programme development. Participants advised that students should avoid cancelling or rearranging visits as this could cause anxiety for the child. Participants suggested the need for flexibility during the visits based on the needs of the child; for example, some children might not participate in the conversation, while others

might enjoy engaging in an activity like drawing pictures. It was perceived that advance information and a photograph of the medical student would reduce possible anxiety in the autistic child.

## DISCUSSION

This study has generated insights on the medical experiences of families with autism. The results of this qualitative study will be used to inform the development of the TfA programme and can be used to inform other such programmes.

While some of the direct examples provided by participants related to GPs and paediatricians, participants perceived that most doctors lack awareness of autism and did not work in partnership with parents of autistic children. Participants were strongly supportive of tailored education to address these concerns. Crucially for the development of TfA, participants in this study reported a willingness to take part directly in medical education and share their lived experience to address this challenge. This suggests that the TfA programme is a feasible aspiration.

Participants in this study highlighted delays in obtaining an autism diagnosis for their child and frustration with the process. A delay of more than 4.5 years between parents detecting their children's developmental issues and receiving a formal autism diagnosis is reported in the literature, and the extent of this diagnostic delay is a predictor of overall parent satisfaction with the diagnostic experience.[27] Despite examples of positive experiences, participants overwhelmingly reported negative experiences relating to diagnosis, and perceived that a lack of autism awareness led to delays in diagnosis.

Participants in this study identified that medical management of comorbidities was often problematic for both parents/carers and the autistic children, leading to differing perceptions of health needs and avoidance of primary care by some participants. This is supported in the wider literature, where it is recognised that comorbidities in autistic children are often misdiagnosed and mismanaged.[28–30]

Mirroring the wider literature, participants felt they were not regarded as experts in dealing with their children's autism or valued when making joint decisions with doctors.[31 32] These findings demonstrate that doctors must work in partnership with patients/carers of autistic children to improve treatment outcomes,[29 30 33] as well as improved parent/carer satisfaction.

The success of the doctor's interactions with autistic people depends on their ability to adapt their behaviour to address the needs of the person.[33] Unsurprisingly, participants in this study valued doctors personalising their approach when communicating with autistic children. However, in order for doctors to adapt their behaviour, they need to be equipped with the necessary attitudes, skills and knowledge to support autistic people.[34]

Participants perceived that TfA would improve student understanding of and approaches towards autistic people. This is supported by research which suggests that this kind of 'real-life' exposure enables students to gain positive attitudes and values that cannot be learnt through textbooks.[35] Longitudinal educational models that offer continuity of contact between student learners and patients/carers can enhance insight into the psychosocial side of caring for patients and understanding of chronic health conditions.[36 37]

This study has three main limitations. First, this study used a convenience sample and therefore lacks diversity in terms of ethnicity and economic status. Our participants were predominantly white, female parents/carers. We were therefore not able to assess the experiences of all groups of carers, which is of concern as it is suggested that autistic children from ethnic backgrounds experience more barriers in the healthcare setting.[38] Additionally, it is not clear whether parents from more diverse backgrounds would be willing to take part in a programme like TfA, potentially limiting exposure to the full range of experiences for medical students. It is evident that further research using a more diverse sample of participants is necessary. Second, this study was undertaken in the south of England, and given the variation in support available geographically for autistic children,[27] further research across a wider geographical area would be beneficial. Finally, lived experience involvement of parents/carers to advise on the design, delivery and analysis would have strengthened the study.

A strength of this study is an improved understanding of the medical care experiences of autistic children and their parents/carers for the purposes of the design and delivery of autism education in undergraduate healthcare curricula. Their positive attitudes towards TfA demonstrates enthusiasm for the active involvement of parents/carers and autistic children in educational delivery.

## CONCLUSION

Despite calls for improved autism knowledge and awareness in the medical profession, this study found that parents/carers of autistic children perceived a generalised lack of understanding of autism in medical settings.

While there is a lack of evidence demonstrating new and innovative ways to enhance autism awareness, education that incorporates lived experience can enhance positive attitudes and values towards chronic conditions. Findings from this study can be used in the design of new initiatives to improve autism awareness in healthcare education. Participants in this study were willing to take part in new programmes, such as TfA to improve understanding of autism in future doctors. A full evaluation of the experience of medical students and parents/carers in TfA will be undertaken on the new programme.

**Acknowledgements** We thank the parents who took part in this study and also Alison Smith and Dr Lisa Butterworth-Salmon for reviewing the interview topic guides.

**Contributors** Conception and design: SD, AW, SB and JW. Data collection: SD and LG. Analysis: YD, YF and SD. Manuscript drafting: YD, SD and YF. All authors read and approved the final manuscript. Reesponsibiliry for overall content as guarantor: SD.

**Funding** This study was funded by Health Education England, Kent, Surrey, Sussex.

**Competing interests** No, there are no competing interests.

**Patient and public involvement** Patients and/or the public were not involved in the design, conduct, reporting or dissemination plans of this research.

**Patient consent for publication** Not applicable.

**Ethics approval** This study involves human participants and was approved by South Central Hampshire B Research Ethics Committee (reference number 19/SC/0041). Informed written consent was obtained from participants included in the study. The participants gave informed consent to participate in the study before taking part.

**Provenance and peer review** Not commissioned; externally peer reviewed.

**Data availability statement** Data are available upon reasonable request. The datasets used and/or analysed during the current study are available from the corresponding author on reasonable request.

**ORCID iD**
Stephanie Daley http://orcid.org/0000-0002-5842-0492

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
