## [Reviewer comments · BMJ Paediatrics Open]

ARTICLE DETAILS

TITLE (PROVISIONAL)	Developing undergraduate autism education for medical students: a qualitative study
AUTHORS	Dhuga, Yasmin Feeney, Yvonne Gallaher , Laura White, Ann Wright , Juliet Banerjee, Sube Daley, Stephanie

VERSION 1 – REVIEW

REVIEWER	Reviewer name: Dr. Georgina Cox Institution and Country: Murdoch Children's Research Institute, Australia Competing interests: None
REVIEW RETURNED	17-Mar-2022

GENERAL COMMENTS	Thank you for the opportunity to review the manuscript titled "Developing undergraduate autism education for medical students: a qualitative study". This is an important area of research and clinical practice that requires attention, in particular a strength of this paper is integrating the views of parents into the program 'Time for Autism'. The sooner we begin to educate professionals about Autism Spectrum Disorder (ASD) the better the outcomes will be for families when they access healthcare services. I have suggested some minor revisions to the manuscript below, that I hope the authors will take on board in order to strengthen the readability of the manuscript. Abstract Background: reword sentence "A new educational program, Time for Autism, designed for medical students has been designed to address this challenge" so design is not repeated. Methods: Include age range of children. Background -provide ref for increasing prevalence and short explanation as to why this may be. -parag 2: reword to clear differentiate between physical and mental health concerns and group accordingly. -Page 5, parag 2: incomplete sentence "...relational learning between the family and the student learns...." -Not covered, but worth considering: ASD can also be present in parents. I wonder how training may also address this, and ways to communicate with parents who also identify as being on the spectrum themselves? Methods Setting and participants: Please clarify sentence "There was no patient or parent involvement in the study"? Parents are being interviewed? Procedure: Please clarify sentence "The majority of interviews took
---

	place at home...with the majority taking place at home” -Why was feedback on findings not sought and why? Would this have strengthened the study and future implications? Discussion -some wider questions to consider: 1. How feasible is the Time for Autism (TfA; consider using acronym throughout also), in terms of pairing up doctors with families? 2. More on the self selection nature of the study and how this might impact results? The issue of low SES and CALD families is touched on briefly. 3. More information on the medical setting. Are participants referring to GPs, paediatricians? Hospital staff? There are a number of settings and teams medical staff work in and more information on this would be helpful.
--	--

REVIEWER	Reviewer name: Dr. Sylvana Mahmic Institution and Country: BURKE ST New South Wales, Australia Competing interests: None
REVIEW RETURNED	22-Mar-2022

GENERAL COMMENTS	Reviewer feedback 18th March 2022 Thank you for the opportunity to read the article submitted to BMJ Paediatrics Open 'Developing undergraduate autism education for medical students: a qualitative study'. The article is suitable for publication in BMJ Paediatrics Open however, needs major revision. Overall, the research topic is very interesting and the Time for Autism training for trainee doctors is an innovative and practical way for them to get to know children with autism in their family context. This practical experience has the potential to shape their views and understanding so that they are better equipped to provide high quality health care to children with autism and their families in the future. I was particularly interested with your finding that parents are willing to contribute their experiences to support the education of trainee doctors and welcome the use of a qualitative approach. My general feedback is that the article requires further development. In addition, thorough editing is required to ensure it meets the standard of a Q1 journal. For example, check that each paragraph has a topic sentence to introduce an idea which is then expanded. Reading aloud will help you identify the typos and unclear sentences. More specifically: 1. Abstract In the Background section, the first few sentences focus on autistic adults. These sentences could be strengthened by rewriting them to make a better connection to your study participants which are children and their families rather than adults. 2. Study design This section was only one sentence which is very brief. It could be strengthened by including references to support your design along with explanation on your rationale for selecting this design. 3. Setting and participants The final sentence is not complete and appears to contradict information in the next procedure section. It firstly states that there was no patient or parent involvement but in the procedure section it states that parents were interviewed at home. I was unsure who the participants were throughout my first reading and had to re-read carefully to determine that the trainee doctors were not a part of the study. 4. Data analysis Fieldwork diaries were completed in addition to the interviews yet there seems to be no reference to this data in the results. 5. Results This starts with two sentences on participants and characteristics
--

	which I suggest would be better in the setting and participants section along with Table 1. This is followed by the three themes which were unclear. The first two themes are very similar, and the topic sentences of each theme seems to be the same that is, participant experiences with doctors. Furthermore, in the diagnosis theme, three sentences start with 'most participants' or 'many participants' which is vague for a sample of 11. There is nothing wrong with this size sample if it was clear in this section that your data supported the theme. However, the table with long sample quotes did not make a clear connection with your results. You could consider integrating selected quotes into the results to help the reader understand the themes better. In theme two, 'parental expectations of doctors', the suggestion that doctors think imaginatively was not clear without supporting quotes from your data. Overall, I was not clear about your findings as the themes were not supported with data. There are some good findings for example, that parents expected that doctors would be willing to involve parents in decision making but these were not elaborated on in detail. In the Time for Autism theme, Visit Considerations, the opening sentence was unclear and had me wondering if the parents did indeed participate or were they just willing to participate. Perhaps it would help the reader if you could consider choosing either parents or participants and then use the selected term consistently. Discussion Your first sentence says that your study generated a novel insight and I had to re-read the results to figure out what you were referring to. Two new ideas seemed to be introduced which I didn't recall from the results (paragraph 3 and 4). I wasn't certain reading the discussion that your results were being discussed. A further limitation is that the study did not take account of the trainee doctors. I would have liked to understand this but since you didn't include them, an explanation would be helpful.
--	--

VERSION 1 – AUTHOR RESPONSE

Comments Our response Editor in Chief Discussion P14 line 20 delete the sentence "To the best of our knowledge, this study is the first to explore the views of parents/carers of autistic children around the design of an autism medical educational programme." Journal policy is for authors to NOT describe their study as the first. We have removed this statement. The English needs improving We apologise for this. The document has been fully reviewed and amendments made Reviewer 1

Abstract Background: reword sentence "A new educational program, Time for Autism, designed for medical students has been designed to address this challenge" so design is not repeated. We have amended this sentence. A new educational programme, Time for Autism, for medical students has been developed to address this challenge

Methods: Include age range of children. This has been added. The ages of the autistic children ranged from 3-17 years

Background Provide ref for increasing prevalence and short explanation as to why this may be. Thank you for point, we have added a reference for the increased prevalence. We have also added a sentence and references about the possible reasons why The cause of increasing prevalence is unknown but likely to be due in part to increased community awareness amongst the general population, including families and teachers and progress in case identification and definition.

Parag 2: reword to clear differentiate between physical and mental health concerns and group accordingly. We have re-worded as follows: including both mental health problems such as anxiety-related disorders (4), sleep disorders (5), as

well as physical health problems such as gastrointestinal conditions (6). Page 5, parag 2: incomplete sentence "...relational learning between the family and the student learns..." Our apologies for this error. We have completed the sentence as follows: Students learn about the lived experience of autism in the context of authentic relationship between the family and the student learners. Not covered, but worth considering: ASD can also be present in parents. I wonder how training may also address this, and ways to communicate with parents who also identify as being on the spectrum themselves? Thank you for this helpful point. Approximately 20% of the parents who have enrolled onto the programme have a diagnosis of autism themselves. We have addressed communication with both adults and children as part of our student teaching. For the manuscript, we have added in the following sentence It is also anticipated that several of the parents taking part in the new programme, will themselves have a diagnosis of autism, further enhancing the learning for medical students. Methods Setting and participants: Please clarify sentence "There was no patient or parent involvement in the study"? Parents are being interviewed? As part of the submission guidance, we were advised to put a statement into the manuscript if there was no lived experience/consumer input into the study design/delivery and analysis As we were in the process of setting up the new programme, we were not in a position to have an advisory group at the stage of setting up this study. We have amended the text to make this clearer. There was no lived experience advisory involvement in this study, although this has since been established for the TfA programme. Procedure: Please clarify sentence "The majority of interviews took place at home...with the majority taking place at home" We apologise for this error. We have amended the text as follows: The majority of interviews took place at home with only the parent/carer present, at a time and venue suitable to participants. Why was feedback on findings not sought and why? Would this have strengthened the study and future implications? As mentioned previously, at the time of undertaking this study we were at a very early stage of development for the TfA programme and did not have an established lived experience advisory group to call upon to help us to interpret the study findings. This has since be rectified. We do agree that this would have strengthened the findings. We have added this as a study limitation. Finally, lived experience involvement to advise on the findings would have strengthened the study. Discussion: How feasible is the Time for Autism (TfA; consider using acronym throughout also), in terms of pairing up doctors with families? Thank you for this suggestion, we have used the acronym TfA throughout Our study (and experience more recently) indicates that parents are willing to take part in medical educational programmes for students. Whilst we did not ask participants directly, we would imagine that there would be willingness to work with doctors, as well of students. However, as this study is about medical students only (as opposed to doctors), we have highlighted parental willingness (and therefore feasibility) in the beginning of the discussion as follows: Crucially, parents of autistic children indicated a willingness to take part in medical education to address this challenge. More on the self selection nature of the study and how this might impact results? The issue of low SES and CALD families is touched on briefly. Thank you for this helpful point. We have expanded the impact of this on the results and TfA programme within the study limitation sections First, this study used a convenience sample, and no attempt to obtain diversity in the sample, in terms of ethnicity and economic status. Our participants were predominantly white, female parents/carers. We were therefore not able to assess the experiences of all groups of carers, which is of concern as it is suggested that autistic children from ethnic backgrounds experience more barriers in the healthcare setting (40). Additionally, it is not clear whether parents from more diverse background would be willing to take part in a programme like TfA, potentially limiting exposure to the full range of experiences for our medical studetns. It is

evident that further research on a more diverse sample of participants is necessary. More information on the medical setting. Are participants referring to GPs, paediatricians? Hospital staff? There are a number of settings and teams medical staff work in and more information on this would be helpful. We sought parental opinions on medical care generally, although some of the responses, particularly related to the diagnosis were more focussed on GPs and Paediatricians. We have therefore amended the text in the manuscripts when specific medical specialities are referred to.

Reviewer 2 In addition, thorough editing is required to ensure it meets the standard of a Q1 journal. For example, check that each paragraph has a topic sentence to introduce an idea which is then expanded. Reading aloud will help you identify the typos and unclear sentences. We apologise for this. The document has been fully reviewed and amendments made.

Abstract In the Background section, the first few sentences focus on autistic adults. These sentences could be strengthened by rewriting them to make a better connection to your study participants which are children and their families rather than adults. Thank you for this point and for the confusion caused. We have reworded both the abstract and the introductory section of the manuscript. We have also added a further reference. We have made it clearer that this paragraph so that it is clearer that the challenges experienced by autistic people, relate to both adults and children, as our programme is intended to teach our students more broadly about autism (not only about the experiences of children). Autistic people, both adults and children are more likely to have additional health conditions compared to the general population (4), including both mental health problems such as anxiety-related disorders (5), sleep disorders (6), as well as physical health problems such as gastrointestinal conditions (7). Autistic adults and children are also at greater risk of health inequalities, and many autistic adults report negative healthcare experiences (8,9). Parents of autistic children describe the healthcare management of their children as inadequate (10). It is widely recognised that poor care experiences, as well as serious concerns about the quality of care provided to autistic adults and children demonstrates a need for improved understanding of autism across the lifespan within the health and social care workforce (11-15). We have made it clearer in the description of TfA programme, that there will be a focus on autism across the lifespan. Supporting teaching activities will focus on autism across the life-span not just in childhood, and it is anticipated that several of the parents taking part in the new programme, will also have a diagnosis of autism, further enhancing the learning for students.

Study design This section was only one sentence which is very brief. It could be strengthened by including references to support your design along with explanation on your rationale for selecting this design. Thank you for this suggestion. We have added the following text: We carried out a qualitative study using individual semi-structured interviews, in order to explore namely parent/carer views about the medical care experiences of their autistic children as well as a new medical education programme in more depth. The study considered the reality of participants' perceptions through an exploration of their experiences and interpretations they attach to them, within a broader social construct (25). This epistemological approach loaned itself to the phenomenological methodology of thematic analysis and is steered by Braun and Clarke's guided approach to thematic analysis.

Setting and participants The final sentence is not complete and appears to contradict information in the next procedure section. It firstly states that there was no patient or parent involvement but in the procedure section it states that parents were interviewed at home. I was unsure who the participants were throughout my first reading and had to re-read carefully to determine that the trainee doctors were not a part of the study. Participants in this study were parents/carers of autistic children. As part of the journal submission guidance, we were instructed to put a statement into the manuscript as to whether or not there was

lived experience/consumer input into the study design/delivery We have amended the text to make this clearer. There was no lived experience advisory involvement in this study, in relation to the study design, delivery and analysis although this has since been established for the TfA programme. We have also added a statement at the end of the manuscript to make clear that future evaluation on trainee doctors will take place once the TfA programme has been delivered. Evaluation of the future impact of TfA on medical student learning about autism will be carried out once the programme has been delivered. Data analysis Fieldwork diaries were completed in addition to the interviews yet there seems to be no reference to this data in the results. Field-work diaries were used to enhance the process of analysis rather than being used as a source of data. We have made this clearer in the section on data analysis Results Starts with two sentences on participants and characteristics which I suggest would be better in the setting and participants section along with Table 1. Thank you for this suggestion, we have moved this table This is followed by the three themes which were unclear. The first two themes are very similar, and the topic sentences of each theme seems to be the same that is, participant experiences with doctors.. Thank you for this very helpful suggestion, we have consolidated these two themes into one theme related to experiences with doctors. Furthermore, in the diagnosis theme, three sentences start with 'most participants' or 'many participants' which is vague for a sample of 11. There is nothing wrong with this size sample if it was clear in this section that your data supported the theme YF That you for this helpful suggestion, we have provided this information. However, the table with long sample quotes did not make a clear connection with your results. You could consider integrating selected quotes into the results to help the reader understand the themes better. The table was used to small word count allowable for the manuscript. We hope that the amendments that we have made to the text, and to the table are helpful in addressing these concerns. In theme two, 'parental expectations of doctors', the suggestion that doctors think imaginatively was not clear without supporting quotes from your data. Thank you for this suggestion, we have tried to illustrate this point more clearly. Participants also suggested that doctors should be willing to be deviant from the norm when assessing or assessing comorbidities in an autistic child, because they may present atypically or the child may be unwilling to engage, for example be willing to assess the child in the family car if they won't come into the practice. Overall, I was not clear about your findings as the themes were not supported with data. There are some good findings for example, that parents expected that doctors would be willing to involve parents in decision making but these were not elaborated on in detail. We have re-worded the text in the findings and reduced the quotes in the tables to make the illustrations clearer. We hope that this addresses this concern. In the Time for Autism theme, Visit Considerations, the opening sentence was unclear and had me wondering if the parents did indeed participate or were they just willing to participate. Perhaps it would help the reader if you could consider choosing either parents or participants and then use the selected term consistently. At the time of this study, the new TfA programme had not started and we were asking parents/carers (our study participants) if they would be willing, in principle to participate, to which they said yes. Since the completion of this study, the TfA programme has commenced and has recruited a large network of families for the programme. We have amended to text to make it clear that the TfA programme had not started at the time of this study. We have also ensure that we use the term Discussion Your first sentence says that your study generated a novel insight and I had to reread the results to figure out what you were referring to. We have removed this statement and have been more modest about the study contribution. Two new ideas seemed to be introduced which I didn't recall from the results (paragraph 3 and 4). I wasn't certain reading the discussion that your results were being

discussed. Thank you for this point, we have more clearly articulated the findings in relation to length of diagnosis and management of comorbidities earlier in the manuscript. We have removed reference to family functioning as this was not identified in the findings. A further limitation is that the study did not take account of the trainee doctors. I would have liked to understand this but since you This study was undertaken to underpin the development of the new Time for Autism (TfA) education programme, specifically to assess the willingness of families didn't include them, an explanation would be helpful. (parents/carers with an autistic child) to volunteer their time. We have only started to deliver TfA to our medical students in this current academic year, and we will be carrying out a formal mixed-methods evaluation for our medical students, eg trainee doctors. Hence why trainee doctors are not interviewed in this study We recognise from the comments from both reviewers that we have not made this distinction clear enough in the manuscript, eg that TfA was not being delivered to our at the stage at which this study was carried out. We have amended the text to make it clearer that TfA is a future aspiration. The incorporation of lived experience to raise awareness and enhance skills towards autism in undergraduate medical students is a key feature of a new autism education programme, Time for Autism programme (TfA) which is being developed at Brighton and Sussex Medical School. TfA will form a mandatory component of the undergraduate medical curriculum, having been modelled on a similar educational programme (19) that successfully enhanced understanding and attitudes of dementia in large cohorts of healthcare students (20,21). In TfA, pairs of medical students will visit a family (parent/carer with an autistic child) on three occasions over one academic year. Students will learn about the lived experience of autism in the context of authentic relationship between the family and the student learners. It is intended that the longitudinal nature of the programme will allow students sufficient time to develop a fuller understanding of how autism affects the child and family over time, further facilitating better understanding of the condition. It is also anticipated that several of the parents taking part in the new programme, will themselves have a diagnosis of autism, further enhancing the learning for medical students. Critical to the establishment of TfA will be the recruitment of volunteer families willing to have medical students visit them at home for the purpose of the new programme. We have also made clear at the end of the manuscript that a future evaluation of the impact of the new programme on medical students/parents will be undertaken. A full evaluation of the experience of medical students and parents/carers in TfA will be undertaken following delivery of the new programme.

VERSION 2 – REVIEW

REVIEWER	Reviewer name: Dr. Georgina Cox Institution and Country: Murdoch Children's Research Institute, Australia Competing interests: None
REVIEW RETURNED	29-May-2022
GENERAL COMMENTS	Thank you to the authors for their detailed responses to all points raised by the reviewers. These changes address the concerns and I now endorse this manuscript for publication. I hope that it benefits the medical community to support medical students (and those later in their training), to partner with those living with autism in the community.

VERSION 2 – AUTHOR RESPONSE

Centre for Dementia Studies Brighton and Sussex Medical School Trafford Centre for Medical Research University of Sussex Falmer East Sussex BN1 9RY 10th June 2022 Dear Dr Bhopti and Professor Choonara Amendments for Manuscript ID bmjpo-2022-001411 - "Developing undergraduate autism education for medical students: a qualitative study" We would like to thank the reviewers for their approval of our manuscript. We can confirm that we have made the final edits as requested below: Comments by Dr Bhopti Response P5 Line 16 - spelling medial should be medical – amended page 5 We apologise for this error. This has been amended on page 5 P8 Line 27 and P9 Line 8 - Gender of researchers need not be mentioned so please take the "female" out from these sentences We had included this information due in keeping with the COREC checklist. 'Female' has been removed on page 7/8 P8 Line 43-44 - Please add a sentence as to why the feedback on findings was not sought We have added a sentence as requested on page 7 of the manuscript. At the time of undertaking this study, the TfA programme was at an early stage of development for the TfA programme and did not have an established lived experience advisory group to call upon to help us to interpret the interview findings. We hope that this meets with your approval Yours sincerely Dr Stephanie Daley Senior Lecturer in Older People's Mental Health & Education